# Opposite Effects of mRNA-Based and Adenovirus-Vectored SARS-CoV-2 Vaccines on Regulatory T Cells: A Pilot Study

**DOI:** 10.3390/biomedicines11020511

**Published:** 2023-02-10

**Authors:** Francesca La Gualana, Francesca Maiorca, Ramona Marrapodi, Francesca Villani, Marzia Miglionico, Stefano Angelo Santini, Fabio Pulcinelli, Laura Gragnani, Silvia Piconese, Massimo Fiorilli, Stefania Basili, Milvia Casato, Lucia Stefanini, Marcella Visentini

**Affiliations:** 1Department of Translational and Precision Medicine, Sapienza University of Rome, Viale dell’Università 37, 00185 Rome, Italy; 2Department of Basic, Clinical, Intensive and Perioperative Biotechnological Sciences, Catholic University, Largo Agostino Gemelli 8, 00168 Rome, Italy; 3Synlab Italia, Via Martiri delle Foibe 1, 20900 Monza, Italy; 4Department of Experimental Medicine, Sapienza University of Rome, Viale dell’Università 37, 00185 Rome, Italy; 5MASVE Interdepartmental Center, Department of Experimental and Clinical Medicine, University of Florence, Largo Brambilla 3, 50134 Florence, Italy; 6Department of Internal Clinical Sciences, Anaesthesiology and Cardiovascular Sciences, Sapienza University of Rome, Viale dell’Università 37, 00185 Rome, Italy

**Keywords:** SARS-CoV-2, vaccines, regulatory T-cells, Treg, tolerance, immunogenicity

## Abstract

New-generation mRNA and adenovirus vectored vaccines against SARS-CoV-2 spike protein are endowed with immunogenic, inflammatory and immunomodulatory properties. Recently, BioNTech developed a noninflammatory tolerogenic mRNA vaccine (MOGm1Ψ) that induces in mice robust expansion of antigen-specific regulatory T (Treg) cells. The Pfizer/BioNTech BNT162b2 mRNA vaccine against SARS-CoV-2 is identical to MOGm1Ψ except for the lipid carrier, which differs for containing lipid nanoparticles rather than lipoplex. Here we report that vaccination with BNT162b2 led to an increase in the frequency and absolute count of CD4^pos^CD25^high^CD127^low^ putative Treg cells; in sharp contrast, vaccination with the adenovirus-vectored ChAdOx1 nCoV-19 vaccine led to a significant decrease of CD4^pos^CD25^high^ cells. This pilot study is very preliminary, suffers from important limitations and, frustratingly, very hardly can be refined in Italy because of the >90% vaccination coverage. Thus, the provocative perspective that BNT162b2 and MOGm1Ψ may share the capacity to promote expansion of Treg cells deserves confirmatory studies in other settings.

## 1. Introduction

Vaccines against SARS-CoV-2 have been rapidly developed after the beginning of the pandemic and represent the most powerful weapon available so far for abating the morbidity and the mortality of COVID-19, which has caused worldwide over 6 million deaths (World Health Organization, https://covid19.who.int, accessed on 18 November 2022). However, a reason of concern is that antibodies against SARS-CoV-2 wane quite rapidly both after vaccination [1,2] and after natural infection [3]. Consequently, while vaccination confers high protection from severe disease, protection from infection and symptomatic disease decreases by 20–50% six to seven months after the second vaccine dose seemingly due to waning of immunity [4]; booster vaccination can, however, prolong protection against mild and severe disease [5].

Several studies have investigated T cell immunity induced by SARS-CoV-2 infection or by mRNA-based and adenovirus vectored vaccines [6]. Interestingly, trials of heterologous vaccination, which admixes priming with one type and boosting with another type of vaccine, reported that priming with the ChAdOx1 nCoV-19 adenovirus vectored vaccine of AstraZeneca and boosting with an mRNA vaccine induced stronger and more durable effector T cell responses than homologous vaccination [7,8,9,10], suggesting higher efficacy of ChAdOx1 nCoV-19 in priming T cell immunity. 

Reactogenicity of SARS-CoV-2 vaccines, which consists of local or systemic symptoms including fever and joint/muscle pain, is likely due to the activation of innate inflammatory responses by vaccine components [10,11], for example by the lipid nanoparticles in mRNA vaccines [12] or by viral DNA in adenovirus-vectored vaccines [13]. Although comparative studies are scarce, it appears that the BNT162b2 mRNA vaccine may be less reactogenic than the ChAdOx1 nCoV-19 adenovirus vectored vaccine [7,14].

Besides the ability to induce antigen-specific responses vaccines mRNA and adenovirus-vectored vaccines are endowed with intrinsic immunomodulatory properties [15,16]. Tolerogenic vaccines are largely meant to induce regulatory T (Treg) cells exerting both antigen-specific and bystander suppression of conventional T cells at sites of inflammation [17]. A breakthrough in the use of vaccines for modulating Treg cells comes from a study in a mouse model of autoimmune encephalitis that recapitulates human multiple sclerosis. The injection of a non-inflammatory mRNA vaccine (MOGm1Ψ) coding for a disease-related autoantigen, developed by BioNTech, resulted in the suppression of disease through the expansion of Treg cells able to execute both antigen-specific and bystander immunosuppression [18]. 

In the context of a pilot study where we compared the immunological changes induced by two vaccines against SARS-CoV-2, we observed that the BNT162b2 mRNA vaccine of Pfizer/BioNTech induced expansion of circulating Treg cells in healthy subjects as well as in patients with mixed cryoglobulinemia (CryoVas), thus recalling the effect of the tolerogenic MOGm1Ψ vaccine in mice [18]. This is interesting because the BNT162b2 and the MOGm1Ψ vaccine are nearly identical except for the lipid component. In contrast, the ChAdOx1 nCoV-19 adenovirus vectored vaccine induced a decrease of Treg cells in healthy subjects. 

## 2. Materials and Methods

### 2.1. Study Design

We investigated three groups of subjects. The first group included 24 patients with hepatitis C virus-associated CryoVas, 75% female and of median age of 79 years (range 50–87); all patients had been treated with direct-acting antiviral agents and were in sustained virologic response. All these patients received two doses of the BNT162b2 vaccine, and 10 of them received a booster dose with the same vaccine 6 months later. This part of the study was approved by the Ethics Committee of the Umberto I Hospital and Sapienza University of Rome (prot. 0486/2021; ClinicalTrials.gov Identifier: NCT04844632) and all participants provided informed consent. 

The second group included 9 healthy donors who received two doses of the BNT162b2 vaccine. There were 4 females and the median age was 29 years (range 27–30). The third group included 16 healthy donors who received the ChAdOx1 nCoV-19 vaccine. There were 6 females (37%) and the median age was 26 years (range 24–40). This part of the study was approved by the Ethics Committee of the Umberto I Hospital and Sapienza University of Rome (Prot. 0470/2021; ClinicalTrials.gov, accessed on 14 April 2021, Identifier: NCT05171959) and all participants provided informed consent. 

### 2.2. Flow Cytometry and Gating Strategy

Flow cytometry studies were performed immediately before the first and 8–14 days after the second vaccine dose; studies were repeated immediately before and 8 days after the booster in CryoVas patients who received a third dose.

Lymphocyte populations were analyzed using a FACSCalibur instrument (Becton-Dickinson, Milan, Italy). Combinations of antibodies conjugated to different fluorochromes (fluorescein, phycoerythrin, allophycocyanin and peridinin-chlorophyll-protein) were used for direct staining. B cell markers analyzed were: CD19, CD20, CD21, CD24, CD27, CD38, IgM; T cell markers were: CD3, CD4, CD8, CD25, CD56, CD127, HLA-DR; all antibodies were from BD Biosciences.

### 2.3. Anti-SARS-CoV-2 Serology

Anti-SARS-CoV-2 spike IgG antibodies were measured using the SARS-CoV-2 IgG II Quant antibody test (Abbott Laboratories, Chicago, IL, USA). Antibody titers are expressed as Binding Antibody Units (BAU)/mL, with a cutoff for positive testing of 7 BAU/mL.

### 2.4. Statistical Analysis

Descriptive data were expressed as median (range) or mean (standard deviation) for continuous variables. Categorical data were compared with the Fisher’s exact test. Continuous variables were compared with the Mann-Whitney test, the Wilcoxon matched-pairs signed rank test or the Kruskal-Wallis test using the GraphPad Prism version 9.1.2 software. A *p*-value of less than 0.05 was considered significant.

## 3. Results

The general aim of this study was to compare the effects of vaccination with the mRNA-based BNT162b2 vaccine and the adenovirus vectored ChAdOx1 nCoV-19 vaccine on the kinetics of the major T and B cell subpopulations in healthy subjects and in patients with CryoVas.

Concerning CryoVas, we measured the distribution of circulating lymphocyte subsets in 24 patients before vaccination and 8–14 days after the second dose of the BNT162b2 vaccine. After vaccination there were no significant changes (Table 1) in the frequency of total CD19+B cells, IgM+CD27^neg^ naïve, IgM+CD27+IgM memory, IgM^neg^CD27+ switched memory or CD24^high^CD38^high^ transitional B cells; also, there was no change in the frequency of the functionally exhausted CD21^low^ B cells that are commonly expanded in CryoVas [19]. T cell subsets were evaluated in 23 patients, and we found no significant changes in the relative frequencies of CD3+, CD4+, CD8+, CD4-CD8- or activated HLA-DR+ T cells, while there was a moderate decrease in the percentage of CD3+CD56+ natural killer T (NKT) cells (Table 1). Strikingly, we observed a robust increase of circulating CD4^pos^CD25^high^ Treg cells; this data, as well as comparative findings in healthy subjects who received either the BNT162b2 or the ChAdOx1 nCoV-19 vaccine, will be described in detail below.

Nine CryoVas patients (37%) failed to produce anti-spike IgG antibodies after the second vaccine dose; the only variable significantly associated with lack of response was an absolute count of circulating B cells of less than 5/µL at vaccination due to recent treatment with rituximab (5/9 vs. 0/15, *p* = 0.003); leukemia-like monoclonal B cell expansion was present in 2/9 non-responders. No correlations could be found between the pre- or post-vaccination frequencies of B and T cell subsets, including Treg cells, and the decline of anti-spike antibody titers, which overall decreased from a mean ± SD of 696 ± 1356 BAU/mL after two doses to 432 ± 1066 BAU/mL before the booster dose (*p* = 0.002).

Treg cells were identified either as CD3^pos^CD4^pos^CD25^high^ cells or as CD3^pos^CD4^pos^CD25^high^CD127^low^ cells, which are nearly homogeneously Foxp3^pos^ [20]. The reason was that the study was initially aimed at an exploratory overview of the changes of lymphocyte subpopulations in patients receiving two different vaccines, and therefore the simplest protocol for identifying Treg cells was initially adopted while CD127 staining was introduced at a later phase. Intracellular staining of Foxp-3 was omitted at this later phase since initial analyses had been done using a four-color flow cytometer, and therefore the addition of Foxp-3 staining would have forced either to change the initial combination of fluorochromes or to use another instrument, thus precluding a reliable pairwise comparison of pre and post vaccination samples. Since the exclusion of patients in which pre-vaccination Treg cell levels had been evaluated only by CD4/CD25 staining would have significantly reduced the sample size, we choose to exploit the CD4^pos^CD25^high^ phenotype for the pairwise comparison of putative Treg cells in pre- and post-vaccination samples. 

When CD25 staining was not available for pre-vaccination samples we adopted the gating strategy illustrated in Figure 1 for comparisons. Electronically gated CD3^pos^CD4^pos^ cells were analyzed for CD25 and CD127 expression, and CD4^pos^CD25^high^CD127^low^ cells were gated; the same CD25 gate used for defining CD4^pos^CD25^high^CD127^low^ cells was then used for comparing the frequency of CD4^pos^CD25^high^ cells in pre- and post-vaccination samples.

A major limitation in enumerating Treg cells solely as CD4^pos^CD25^high^ cells is that this population also includes activated T cells. Thus, we took advantage of the samples stained for CD4, CD25 and CD127 to calculate the relative frequency of activated T cells (CD25^high^CD127^pos^) and of Treg cells (CD25^high^CD127^low^) among CD4^pos^CD25^high^ cells (Figure 1A). On average, CD25^high^CD127^pos^ activated T cells represented about one fifth of CD4^pos^CD25^high^ cells (mean ± SD, 21.2 ± 12.2%), and their frequency did not differ significantly in pre- and post-vaccination samples. Although we fully acknowledge the limits of this immunophenotyping strategy, for simplicity we will hereafter refer to CD4^pos^CD25^high^ T cells as Treg cells. 

Before vaccination, the frequency among CD4+ T cells and the absolute count of circulating Treg cells were significantly reduced in CryoVas patients compared to healthy donors (Figure 2A,B). A reduced frequency of Treg cells in patients with CryoVas compared to healthy donors and to patients with different inflammatory disorders was previously reported [21]. After the second dose of the BNT162b2 vaccine there was a significant increase in the percentage (Figure 2C) and in the absolute count (Figure 2D) of Treg cells in CryoVas patients. Ten CryoVas patients received a booster dose of BNT162b2 vaccine 6 months after the second dose. Although the differences in pre- and post-vaccination Treg cell levels did not reach statistical significance, possibly owing to the small sample, both the frequency (Figure 2E) and the absolute count (Figure 2F) of Treg cells were significantly higher after the booster dose compared to pre-vaccination values, suggesting an additive effect of boosting with BNT162b2 on Treg cells expansion. 

Post-vaccination changes in T cell subsets were investigated in 16 healthy donors who received the ChAdOx1 nCoV-19 adenovirus vectored vaccine. Like with the BNT162b2 vaccine, no significant changes in the proportions of major T cell subsets were observed (not shown). However, in sharp contrast with the observations in subjects receiving the BNT162b2 vaccine, after vaccination with ChAdOx1 nCoV-19 there was a significant decrease in the frequency (Figure 2G) and in the absolute number (Figure 2H) of circulating CD4^pos^CD25^high^ Treg cells.

The CD25 mean fluorescence intensity (MFI) of Treg cells did not vary after vaccination either in CyoVas patients who received BNT162b2 or in healthy subjects who received ChAdOx1 nCoV-19 (Figure 2I,J), further suggesting that the increased number observed after the vaccination with BNT162b2 was due to the expansion of canonical Treg cells.

## 4. Discussion

Our preliminary findings suggest that the mRNA based BNT162b2 vaccine induces an increase of Treg cells whereas the adenovirus-vectored ChAdOx1 nCoV-19 vaccine is associated to a decrease in the frequency of Treg cells. This might not be totally surprising, since BNT162b2 is nearly identical, except for the lipid component, to the tolerogenic MOGm1Ψ vaccine that induces a robust expansion of antigen-specific Treg cells in mice [18].

To our knowledge, the dynamics of Treg cells after vaccination with BNT162b2, ChAdOx1 nCoV-19 or other SARS-CoV-2 vaccines has been poorly investigated. Neidleman et al. [22] used CyTOF to phenotype T cell populations in 11 subjects vaccinated with either BNT162b2 (n = 7) or the mRNA-1273 vaccine of Moderna; although the authors state that there were no changes in the frequency of SARS-CoV-2-specific Treg cells, the data displayed show a trend toward and increase after the first and second dose, which may not have reached statistical significance because of the small number of subjects. Also, T cells derived from subjects vaccinated with BNT162b2 or mRNA-1273 produce, when cultured in the presence of spike protein peptides, IFN-γ and IL2 but also abundant IL10 seemingly released by activated Treg cells [23].

Changes in circulating Treg cells have been investigated after vaccinations against pathogens other than SARS-CoV-2. For example, the live attenuated yellow fever vaccine led to a decrease of resting and an increase of activated Treg cells, hepatitis B vaccine increased the frequency of Treg cells, while adjuvanted or non-adjuvanted influenza vaccines had no effect [24]. In contrast with the findings in healthy subjects [24], Salemi and colleagues [25] observed an increase of Treg cells in patients with rheumatoid arthritis treated with TNF-alpha blockers after vaccination with a non-adjuvanted influenza vaccine. Thus, the effects of different vaccines on Treg cell dynamics may depend on the properties of the vaccine as well as on the immunological characteristics of the recipient. Furthermore, the simultaneous administration of multiple vaccines [26] may trigger complex immunologic milieux where cytokines, B cells and possibly Treg cells could contribute to either the enhancement or the dampening of immunologic responses. 

The results of our pilot study must be taken with great caution because of important limitations (see below); nevertheless, these observations may fit with current knowledge about mRNA and adenovirus-vectored vaccines. Our findings suggest that BNT162b2 and the noninflammatory MOGm1Ψ vaccine [18] might share the capacity to induce expansion of suppressive Treg cells. These vaccines, both developed at BioNTech, incorporate a chemically modified nucleoside (1-methylpseudouridin, m1Ψ) that reduces the proinflammatory activity of mRNA by preventing the activation of TLR7, TLR8 and other innate immune sensors [27]. It is surmised that the difference in immunogenicity and tolerogenicity of these vaccines depends on the different encapsulating lipid formulations [28,29], since the BNT162b2 vaccine is carried by lipid nanoparticles (LNPs) containing the ALC-0315 ionizable lipid [30] whereas MOGm1Ψ is encapsulated in a reputedly noninflammatory cationic lipoplex (LPX) originally designed to target the vaccine to the spleen rather than to the lungs [31]. Alameh and colleagues [32] reported that a LNP formulation containing ionizable lipids (different from the LNP of BNT162b2) was a built-in adjuvant of mRNA and protein vaccines by inducing in mice IL-6 and consequently T follicular helper cell responses; they also reported that this LNP did not induce Treg cell responses, although this was shown only with a LNP carrying a protein antigen and was not tested with a mRNA vaccine. A subsequent study [33] showed that the BNT162b2 vaccine-induced IFN-γ-producing CD8^pos^ T cells responses through activation of type I interferon responses induced by MDA-5, a receptor for double-stranded RNA; this suggested that the m1ΨmRNA of BNT162b2 is itself a built-in adjuvant at least in mice. Another study [34] compared in mice the effects of LPX with those of two LNPs, including the LNP containing the SM-102 ionizable lipid used in the Moderna mRNA-1273 vaccine against COVID-19. In this study it was shown that m1Ψ-modified mRNA was non-immunostimulatory (in terms of IL-1 production) when formulated in LPX, potently stimulatory when formulated in LNP SM-102, and only weakly stimulatory when formulated in LNP MC3 nonionizable lipids. In addition, the vaccine formulated in LNP SM-102 upregulated in both in humans and in mice IL-1 receptor antagonist (IL-1ra), a cytokine that blocks the binding of IL-1 to its receptor; however, this resulted in an anti-inflammatory effect only in mice because of their higher constitutive levels of IL1-ra in serum. Altogether, these studies provide a complex scenario in which reactogenicity, immunogenicity and possibly tolerogenicity of m1ΨmRNA vaccines may be context dependent, and further highlight that the results in mice might not faithfully reflect the human immune system response [29]. 

Assuming, again with the due caution, that the BNT162b2 mRNA vaccine expands whereas the ChAdOx1 nCoV-19 adenovirus-vectored vaccine dampens Treg cells, then an impact on vaccine immunogenicity could be predicted since the Treg cells involved would in principle be antigen-specific [18]. Indeed, several studies suggest superiority of adenovirus vectored vaccines compared to mRNA vaccines in inducing strong and durable T cell responses [7,8,9,10,35,36,37], coherently with an opposite effect on Treg cells. A study in elderly subjects [35] reported more frequent and more robust responses of IFN-γ producing CD4+ and CD8+ T-cells (*p* < 0.0001) after a single vaccination with ChAdOx1 nCoV-19 rather than with BNT162b2. Also, studies in subjects receiving heterologous vaccinations concur that priming with ChAdOx1 nCoV-19 and boosting with an mRNA vaccine induces stronger and more durable effector T cell responses than homologous vaccination with either type of vaccine [7,8,36,37]. The Com-COV study [9] compared heterologous versus homologous prime-boost schedules and reported that the number of circulating IFN-γ secreting T cells specific to spike protein were significantly higher with ChAd Ox1 nCoV-19 than with BNT162b2 at 14 days (*p* < 0.0001) and 28 days (*p* < 0.0001) after prime vaccination. The subsequent Com-COV2 study [10] confirmed a higher frequency of IFN-γ-secreting spike specific T cells after priming with ChAdOx1 nCoV-19 rather than with BNT162b2. In addition, this study showed that priming with ChAdOx1 nCoV-19 produced higher final T cell responses than any other combination, with the highest response observed after priming with ChAdOx1 nCoV-19 and boosting with the Novavax peptide-based adjuvanted vaccine and the lowest after priming with BNT162b2 and boosting with Novavax; these findings were interpreted as suggesting a peculiar efficacy of the ChAdOx1 nCoV-19 vaccine in priming T cells for subsequent stimulation by protein antigens. 

Studies with anticancer mRNA vaccines provide clues about a role for Treg cells in dampening anti-tumor efficacy and about ways to manipulate Treg responses [38]. For example, TriMix, a construct containing immunostimulatory mRNAs encoding for CD40 ligand, CD70 and TLR4, if associated to mRNA vaccines leads to a shift from the generation of Treg cells to the generation of Th-1 cells [39] and to an increase of anti-tumor activity of anti-melanoma mRNA vaccine in preclinical and clinical studies [40,41]. 

## 5. Limitations of the Study

Our pilot study suffers from important limitations that make the results highly preliminary and solely suggestive. A major limitation is that the very basic assay identifying Treg cells as CD4^pos^CD25^high^(CD127^low^) T cells is anything but accurate [42,43]. Nevertheless, in earlier studies even the very basic CD4^pos^CD25^high^ assay proved to be reasonably reliable in clinical settings [21,25,44,45,46]; as an example, the claim that Treg cells are reduced in CryoVas patients was initially based on the enumeration of CD4^pos^CD25^high^ cells [21] and was subsequently confirmed by Foxp3 staining [47]. CD127 staining was omitted in the initial phase of the study, and this forced us to eventually adopt the CD4^pos^CD25^high^ phenotype for comparisons. A major problem with this approach is that the CD4^pos^CD25^high^ population also contains CD127^pos^ activated T cells. Among the subjects tested in this study, CD25^high^CD127^pos^ activated T cells represented ~20% of CD4^pos^CD25^high^ cells and that their frequency did not vary significantly after vaccination. Furthermore, it is difficult to explain why the BNT162b2 mRNA vaccine should cause an expansion of activated CD4 T cells while the adenovirus-vectored ChAdOx1 nCoV-19 vaccine, which is more inflammatory and induces more robust effector T cells responses [35], should cause their reduction.

Another important limitation is that we compared the effects of ChAdOx1 nCoV-19 in young and healthy subjects with those of BNT162b2 in older subjects affected by an immunological disease known to be associated with low Treg cell counts. As it has been shown that Treg cells increase in CryoVas patients after treatment with low-dose interleukin-2 [48], it is possible that their increase of after BNT162b2 could be due to the production of this cytokine in response to vaccination, 

## 6. Conclusions

Our preliminary observations, although burdened by important limitations, raise the provocative perspective that the BNT162b2 vaccine might be endowed, at least to some extent, with the capacity to drive expansion of antigen-specific Treg cells analogous to that of the tolerogenic MOGm1Ψ vaccine. Our findings also suggest that the adenovirus-vectored ChAdOx1 nCoV-19 vaccine might have an opposite effect on Treg cell dynamics. It must be stressed that our results are highly preliminary and need confirmation. The advanced stage of the COVID-19 vaccination campaign precludes confirmatory studies in Italy; thus, it is hopeful that state-of-the-art analysis of Treg cell dynamics in SARS-CoV-2 vaccinees will be carried out in other settings. Also, studies in mice directly comparing the effects of MOGm1Ψ and BNT162b2 on Treg cells are advisable.

## Figures and Tables

**Figure 1 biomedicines-11-00511-f001:**
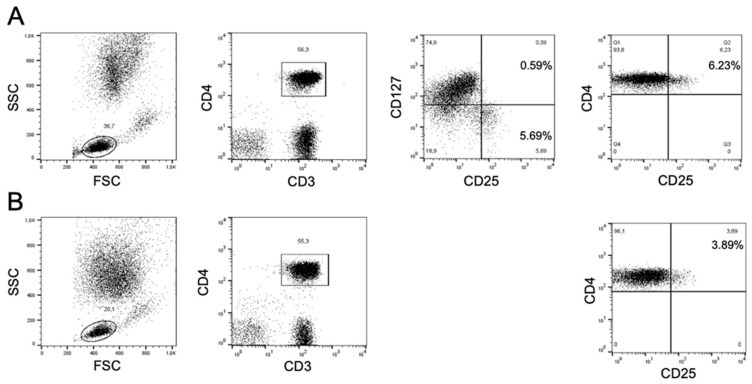
General gating strategy for the pairwise comparison of CD4^pos^CD25^high^ in pre- and post-vaccination samples. (**A**) In this representative case, the post-vaccination sample is stained with antibodies to CD3, CD4, CD25 and CD127; electronically gated CD3^pos^CD4^pos^ cells are analyzed for CD25 and CD127 expression and gates are set to define CD25^high^CD127^low^ (5.69% of cells) and CD25^high^CD127^pos^ (0.59% of cells) populations; the setting of the *x*-axis adopted for defining CD25 positive fluorescence (phycoerythrin) is then used to calculate the total fraction of CD4^pos^CD25^high^ cells in this sample as well as (**B**) in the pre-vaccination sample in which CD127 staining was not done and CD25 was stained with the same phycoerythrin-conjugated antibody.

**Figure 2 biomedicines-11-00511-f002:**
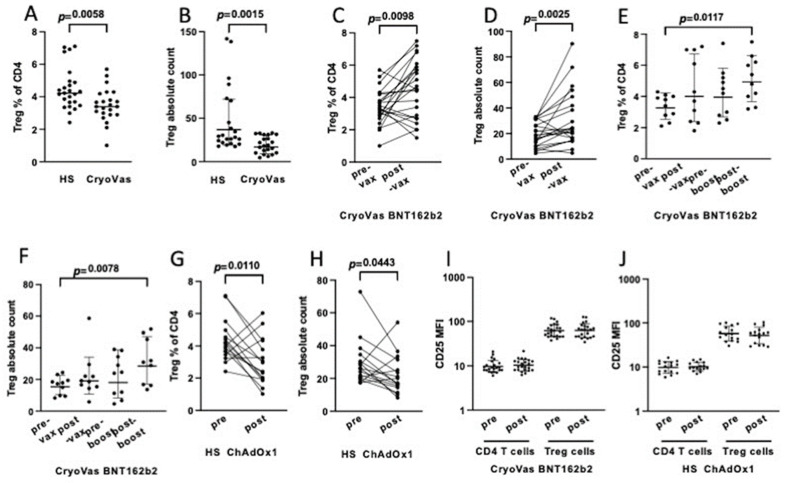
The BNT162b2 mRNA vaccine drives an increase and the ChAd Ox1 nCoV-19 adenovirus-vectored vaccine a decrease of CD4^pos^CD25^high^ Treg cells. (**A**) The frequency among CD4^pos^ T cells and (**B**) the absolute count of Treg cells are significantly reduced in patients with CryoVas compared to healthy subjects (HS). (**C**) The frequency and (**D**) the absolute count of Treg cells increase in CryoVas patients after the second dose of BNT162b2. Findings in 10 CryoVas patients who received a booster dose of BNT162b2 mRNA vaccine 6 months after the second dose show that (**E**) the frequency and (**F**) the absolute count of Treg cells increase significantly after booster vaccination compared to pre-vaccination (pre-vax); none of the other comparisons reaches statistical significance. (**G**) The frequency and (**H**) the absolute count of Treg cells decrease in healthy subjects after the second dose of ChAd Ox1 nCoV-19 vaccine. (**I**,**J**) CD25 mean fluorescence intensity (MFI) in CD4^pos^CD25^neg^ T cells and in CD4^pos^CD25^neg^ Treg cells, before and after vaccination, in (**I**) CryoVas patients vaccinated with BNT162b2 and (**J**) in healthy subjects vaccinated with ChAdOx1 nCoV-19.

**Table 1 biomedicines-11-00511-t001:** Changes of lymphocyte subpopulations after two doses of BNT162b2 vaccine in patients with CryoVas. Data are expressed as mean ± SD; NS, not statistically significant.

Markers	Subset	Before Vaccine	After Vaccine	*p*-Value
B cells				
CD19^+^ (% of lymphocytes)	Total B cells	10.7 ± 17.1	11.6 ± 18.5	NS
CD19^+^IgM^+^CD27^−^ (% of CD19^+^)	Naive	59.1 ± 27	61.1 ± 28.4	NS
CD19^+^IgM^+^CD27^+^ (% of CD19^+^)	IgM memory	14.8 ± 16.8	14.5 ± 17.6	NS
CD19^+^IgM^−^CD27^+^ (% of CD19^+^)	Switched	16.9 ± 14.4	17.5 ± 12.8	NS
CD19^+^CD24^high^CD38^high^ (% of CD19^+^)	Transitional	2.2 ± 2	2.3 ± 3.1	NS
CD19^+^CD21^low^ (% of CD19^+^)	Exhausted	14.7 ± 9.6	14.2 ± 12.9	NS
T cells				
CD3^+^ (% of lymphocytes)	Total T cells	70.6 ± 14	72.5 ± 10.7	NS
CD4^+^ (% of lymphocytes)	Helper/Treg	40.5 ± 10.4	41.1 ± 10.6	NS
CD8^+^ (% of lymphocytes)	Cytotoxic	28.3 ± 11.9	29.4 ± 10.3	NS
CD3^+^CD4^−^CD8^−^ (% of lymphocytes)	Double negative	2.1 ± 2.6	2.3 ± 2	NS
CD3^+^HLA-DR^+^ (% of CD3^+^)	Activated	19.6 ± 18.3	21.5 ± 17.9	NS
CD3^+^CD56^+^ (% of CD3^+^)	Natural killer T	7 ± 8.6	5.5 ± 6.1	0.041
CD4^+^CD25^+^CD127^low^ (% of CD4^+^)	Tregs	3.4 ± 1	4.9 ± 2.1	0.006
NK cells	Natural killer	11.1 ± 4.8	10.6 ± 5	NS

## Data Availability

All data generated or analyzed during this study are included in this published article.

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
