# Peer review of "Opposite Effects of mRNA-Based and Adenovirus-Vectored SARS-CoV-2 Vaccines on Regulatory T Cells: A Pilot Study"

_biomedicines, 2023, doi:10.3390/biomedicines11020511_

Round 1

Reviewer 1 Report

The manuscript submitted by La Gualana entitled "Opposite effects of mRNA-based and adenovirus-vectored SARS-CoV-2 vaccines on regulatory T cells: a pilot study" to the SI "Clinical Immunology in Italy, with Special Emphasis to Primary and Acquired Immunodeficiencies: A Commemorative Issue in Honor of Prof. Fernando Aiuti" of the Biomedicines Journal (MDPI Group) aims to compare the effects of vaccination with the mRNA-based BNT162b2 vaccine and the adenovirus vectored ChAdOx1 nCoV-19 vaccine on the kinetics of the T and B cell subpopulations in healthy subjects and in patients
with CryoVas.

In the opinion of this reviewer, the manuscript is very well-written, showing a good introduction with a nice and adequated numebr of references. The M&M section is clear, allowing the reader to repeat the experiments. The results and the discussion are also very well constructed and build in a very solid results.

However, the following aspects need to be improved before the acceptance of this manuscript for publication:

1) In the M&M the results presented should be transfered to the results section, i.e. the text and the figure 1.

2) In the images of the figure 2, the "p" should be added before the corresponded value

3) In the discussion section, the authors should speculate a bit more about the possible immune responses in humans if the protection will be achieved by using other commercial vaccines.

Author Response

We wish first to thank the reviewers for the appropriate and constructive comments on our work. We tried at our best to answer to their comments as outlined in the following point-by-point reply. Changes are denoted by red color of the text.

Reviewer: The manuscript submitted by La Gualana entitled "Opposite effects of mRNA-based and adenovirus-vectored SARS-CoV-2 vaccines on regulatory T cells: a pilot study" to the SI "Clinical Immunology in Italy, with Special Emphasis to Primary and Acquired Immunodeficiencies: A Commemorative Issue in Honor of Prof. Fernando Aiuti" of the Biomedicines Journal (MDPI Group) aims to compare the effects of vaccination with the mRNA-based BNT162b2 vaccine and the adenovirus vectored ChAdOx1 nCoV-19 vaccine on the kinetics of the T and B cell subpopulations in healthy subjects and in patients 
with CryoVas. 

In the opinion of this reviewer, the manuscript is very well-written, showing a good introduction with a nice and adequated numebr of references. The M&M section is clear, allowing the reader to repeat the experiments. The results and the discussion are also very well constructed and build in a very solid results.However, the following aspects need to be improved before the acceptance of this manuscript for publication:

Comment 1: In the M&M the results presented should be transfered to the results section, i.e. the text and the figure 1.

Reply to comment 1: we moved the indicated part of the M&M and the relevant figure to the Results (page 4, lines 161-201 of the revised manuscript).

Comment 2: In the images of the figure 2, the "p" should be added before the corresponded value.

Reply to comment 2: we added the “p” before the p-values in figure 2.

Comment 3: In the discussion section, the authors should speculate a bit more about the possible immune responses in humans if the protection will be achieved by using other commercial vaccines.

Reply to comment 3: we added further comments and speculation about the issue raised by the reviewer (pager 7, lines 253-264 of the revised manuscript); a new reference has also been added (#26 in the revised manuscript).

Reviewer 2 Report

The article represents a limited pilot study performed in a group of individuals with different conditions and ages. Even though the aim of the study is clear and important, the experimental design is not appropriate. Independently of CD4/CD25/CD127 analysis, which is partially due to the lack of appropriate CD4/CD25/Foxp3 staining, there is no analysis of CD154 as a marker of activation of that specific subpopulation that represents TH1, IFN gamma population, the conclusions of the study are limited. Moreover, as stated by the authors, the old patients treated with rituximab can not be compared in the group; they will not respond. There is also an inappropriate presentation of the data in figure 2; only the absolute count of Treg is important based on the heterogeneous population analyzed. Thus, part A should be modified, B eliminated, as well as F and D, E and H should contain absolute values, and % are misleading. If the authors want to stress the difference in fluorescence intensity, the MFI should be added in the supplement section. The discussion should be enhanced. A section specifying limitations should be included. 

Author Response

We wish first to thank the reviewers for the appropriate and constructive comments on our work. We tried at our best to answer to their comments as outlined in the following point-by-point reply. Changes are denoted by red color of the text.

The article represents a limited pilot study performed in a group of individuals with different conditions and ages. Even though the aim of the study is clear and important, the experimental design is not appropriate. Independently of CD4/CD25/CD127 analysis, which is partially due to the lack of appropriate CD4/CD25/Foxp3 staining, there is no analysis of CD154 as a marker of activation of that specific subpopulation that represents TH1, IFN gamma population, the conclusions of the study are limited. Moreover, as stated by the authors, the old patients treated with rituximab can not be compared in the group; they will not respond. There is also an inappropriate presentation of the data in figure 2; only the absolute count of Treg is important based on the heterogeneous population analyzed. Thus, part A should be modified, B eliminated, as well as F and D, E and H should contain absolute values, and % are misleading. If the authors want to stress the difference in fluorescence intensity, the MFI should be added in the supplement section. The discussion should be enhanced. A section specifying limitations should be included. 

Comment 1: “….there is no analysis of CD154 as a marker of activation of that specific subpopulation that represents TH1, IFN gamma population, the conclusions of the study are limited.”

Reply to comment 1: unfortunately, we did not investigate CD154 (CD40L) to identify activated CD4 T cells; however, in the revised manuscript we address this issue by discussing our data on CD4+CD25+CD127low Treg cells and CD4+CD25+CD127pos activated T cells indicating that the latter do not change significantly after vaccination (page 7 lines 273-276). Anyway, we agree that the conclusions of our study are limited and need confirmation.

Comment 2: “There is also an inappropriate presentation of the data in figure 2; only the absolute count of Treg is important based on the heterogeneous population analyzed. Thus, part A should be modified, B eliminated, as well as F and D, E and H should contain absolute values, and % are misleading.”

Reply to comment 2: we fully agree that the absolute count is more representative of Treg cell dynamics than their percentage among CD4 T cells, and therefore we reported that absolute counts for all comparisons; nevertheless, since the relative frequency of Treg cells is often quoted (e.g. ref. 27 of the revised manuscript) we would propose to keep the figures reporting the corresponding frequencies as shown in the revised figure 2. We omitted from this revised figure the panel H present in the original figure; this panel originally reported the frequency of Treg cells in a small number of healthy subjects vaccinated with tBNT162b2, but since we did not have the Treg absolute counts for some of them (blood cell counts not done) we preferred to withdraw this partial data.

Comment 3: “A section specifying limitations should be included.”

Reply to comment 3: we added a section in which the limitations of our study were more clearly outlined (Discussion page 7, lines 265-289, and a small comment in the Conclusions page 9 lines 360-361).

Round 2

Reviewer 1 Report

The authors positively answered to all raised questions

Author Response

We thank the reviewer for the positive comments.

Reviewer 2 Report

The manuscript has been improved with the text added. However, the authors did not modify the figure as requested and the did not include the MFI of the data as requested in the supplementary files. Without the addition of the data, the paper can not be accepted.

The limitations should be in a section so it is clear for the reader.

Some new sentences require grammatical correction

Author Response

We are sorry that the changes to figure 2 did not meet the reviewer’s request; actually, we thought that it was asked to add the absolute counts of Treg cells where they were missing, and we did not eliminate the panels reporting percentages just for completeness; if this was the request we can do it in a moment.

Concerning the displaying of MIF values, we apologize for having missed it in the last version; now we have included in figure 2 (we thought that there was more immediately visible for the reader than in a supplementary figure) 2 novel panels describing the CD25 MIF values in whole CD4 cells and in Treg cells before and after vaccinations; a comment on this data is on page 6 lines 224-227. We hope that we correctly addressed the reviewer’s request.

We have created a section entitled “Limitations of the study” (#5, from page 8 line 339 to page 9 line 360).

We corrected the grammar of some of the sentences.

New changes are evidenced in blue.